# Lipidomics of Caco-2 Cells Under Simulated Microgravity Conditions

**DOI:** 10.3390/ijms252312638

**Published:** 2024-11-25

**Authors:** Giulia Tolle, Gabriele Serreli, Monica Deiana, Loredana Moi, Patrizia Zavattari, Antonella Pantaleo, Cristina Manis, Mohammed Amine El Faqir, Pierluigi Caboni

**Affiliations:** 1Department of Life and Environmental Sciences, University of Cagliari, 09042 Monserrato, Italy; g.tolle@studenti.unica.it (G.T.); cristina.manis@unica.it (C.M.); 2Unit of Experimental Pathology, Department of Biomedical Sciences, University of Cagliari, 09042 Monserrato, Italy; gabriele.serreli@unica.it (G.S.); monica.deiana@unica.it (M.D.); 3Unit of Biology and Genetics, Department of Biomedical Sciences, University of Cagliari, 09042 Monserrato, Italy; lorymoi@gmail.com (L.M.); pzavattari@unica.it (P.Z.); 4Department of Biomedical Sciences, University of Sassari, Viale San Pietro, 07100 Sassari, Italy; apantaleo@uniss.it (A.P.); m.elfaqir@phd.uniss.it (M.A.E.F.)

**Keywords:** sphingolipids, spaceflights, ceramide signaling, sphingomyelins

## Abstract

Microgravity may profoundly impact the cardiovascular system, skeletal muscle system, and immune system of astronauts. At the cellular level, microgravity may also affect cell proliferation, differentiation, and growth, as well as lipid metabolism. In this work, we investigated lipid changes in Caco-2 cells cultured in a clinostat for 24 h under simulated microgravity conditions (SMC). Complex lipids were measured using a UHPLC-QTOF/MS platform, and the data were subjected to multivariate analysis. Under SMC, levels of ceramides Cer 18:0;O2/16:0, Cer 18:1;O2/16:0, Cer 18:1; O2/22:0, Cer 18:1;O2/24:0, and Cer 18:2;O2/24:0 were found to be upregulated, while sphingomyelins SM 16:1;O2/16:0, SM 16:1;O2/18:1, SM 18:1;O2/24:0, and SM 18:2;O2/24:0 were found to be downregulated. On the other hand, considering that sphingolipids are involved in the process of inflammation, we also treated Caco-2 cells with dextran sodium sulfate (DSS) to induce cell inflammation and lipopolysaccharide (LPS) to induce cell immune responses. As a result, we observed similar lipid dysregulation, indicating that SMC may exert a condition similar to inflammation. Our lipidomics strategy provides new insights into the altered metabolic pathway of ceramides and sphingomyelins of Caco-2 cells under SMC.

## 1. Introduction

Space is considered a severe environment, and long-duration spaceflights strongly affect the human body. Astronauts, during their missions, experience the effects of several stressors such as exposure to solar and ultraviolet radiation, microgravity, isolation, and inadequate sleep and nutrition [1,2]. These factors may lead to adverse health effects on astronauts, which is why investigations of altered biomarkers during spaceflights or in simulated microgravity conditions (SMC) are becoming of paramount importance. Moreover, microgravity induces changes in intracellular signaling, metabolism, and cell communication [3,4,5,6,7]. To study such effects, at the cellular level, 3D clinostats are widely used to simulate weightlessness [8]. A clinostat can simulate microgravity by counteracting the effects of gravity through continuous motion, enabling the study of organism growth and development without directional gravity cues. It achieves this by rotating samples around one or more axes at a steady pace, which disrupts the organisms’ ability to perceive a consistent gravitational force in any direction. Simulated microgravity experiments often involve cell cultures to explore metabolic pathways such as cellular signaling, innate immune responses, and osteogenesis [9,10].

In this context, lipidomics, i.e., the analysis of complex lipids, can be efficiently used to study changes in cell lipid metabolism in an organic extract of a biological sample [11,12].

The human intestinal adenocarcinoma cell lines Caco-2 were previously cultured in SMC to evaluate their performance under different in vitro conditions [13]. Caco-2 cells exhibit both the functional and morphological characteristics of mature absorptive enterocytes, including a brush border layer similar to that found in the small intestine [14]. Consequently, they are widely used as a cell model for the intestinal epithelial barrier [15,16]. The intestinal barrier plays a crucial role in health and disease by facilitating nutrient absorption and preventing the entry of pathogens and harmful chemicals. Moreover, understanding the intestinal barrier modifications is essential for studying the physiology of the gastrointestinal tract and the pathogenesis of inflammatory bowel diseases (IBD), such as Crohn’s disease and ulcerative colitis [17].

Recently, La Barbera et al. reported a proteomic approach and a bioluminescent reporter gene assay to study the effect of simulated microgravity on Caco-2 cells [18]. After 48 and 72 h under microgravity, these authors identified 38 and 26 proteins, respectively, that were differentially involved in regulation, cellular and metabolic processes, and localization. Additionally, bioluminescent reporter gene assays showed lower NF-κB basal activation, corroborating the hypothesis of reduced immunity under microgravity conditions [18]. Alvarez et al. observed, in microgravity conditions, that the intestinal cancer cells HT-29, when cultured on microcarrier beads, showed a defect in the epithelial barrier function [19]. Moreover, Akinsuyi et al. showed, supported by transcriptomic and metagenomic analysis of human and mice samples, that astronauts during space missions may experience the onset of leaky gut [20]. Furthermore, physiologic stressors during spaceflights increase intestinal mucosal paracellular permeability, enhancing paracellular transport of bacteria and bacterial toxins from the lumen to the submucosa into systemic circulation, provoking systemic inflammation [21]. Recently, Yang et al. elegantly reviewed the effects of microgravity on the human digestive system [22].

This study aimed to evaluate the effects of simulated microgravity on the lipidome of Caco-2 cells cultured in a 3D clinostat, with a focus on identifying changes in complex lipid levels. Additionally, under simulated microgravity conditions, we investigated lipid changes in Caco-2 cells when exposed to the pro-inflammatory dextran sodium sulfate (DSS) and lipopolysaccharide (LPS). To achieve this, we performed an untargeted lipidomics analysis using a UHPLC-QTOF/MS platform combined with multivariate analysis (MVA) to identify variations in complex lipids in response to microgravity conditions.

## 2. Results and Discussion

### 2.1. Simulated Microgravity Experiment

Caco-2 cells were placed in a 3D clinostat, and samples were collected at 24 h. After sample extraction, high-resolution mass spectrometric data were acquired, and differences in the levels of complex lipids were explored. Based on MS/MS experiments, we were able to annotate complex lipids such as three lysophosphatidylcholines (LPC), nine phosphatidylcholines (PC), four phosphatidylserines (PS), nine phosphatidylethanolamines (PE), eight ceramides (Cer), nine sphingomyelins (SM), eight acylcarnitines (ACar), two glucosylceramides (Glc-Cer) and twenty triacylglycerols (TG). Discriminant metabolites between Caco-2 cells controls and those placed in the clinostat were obtained from the OPLS-DA multivariate analysis. In Table 1, we report discriminant lipids (VIP > 1 and *p* < 0.05) along with their attribution, mass spectrometric characteristics, and their up- or downregulation in SMC after 24 h. Under SMC, phosphatidylcholines PC 16:0/16:1, PC 16:1/16:1, PC 16:1/18:0, PC 16:1/18:1, PC 15:0/20:4, PC 18:0/18:2, PC 18:0/18:3, and PC 18:0/20:1 were found to be upregulated together with their hydrolysis products, i.e., LPC 16:0, LPC 18:0, and LPC 18:1. From a metabolic point of view, PCs are a major component of Caco-2 cell membranes, contributing to membrane fluidity, integrity, and the formation of lipid bilayers, which are crucial for maintaining the barrier function of the intestinal epithelium. PCs also act as precursors of sphingomyelin and LPCs. PCs are also involved in cell signaling pathways that regulate cell growth, differentiation, and apoptosis [23]. Our findings revealed a process of cellular membrane remodeling occurring under microgravity conditions in accordance with Po et al., 2019. These authors were able to observe a reversible morphology switch in cells under SMC [24].

In addition to phospholipids, sphingolipids such as ceramides (Cer) and sphingomyelins (SM) act as key bioactive molecules to control critical cellular functions, such as cell cycle, aging, apoptosis, cell migration, and inflammation [25]. Ceramides are important components of cell membranes, particularly within the lipid bilayer. They contribute to membrane rigidity and are involved in the formation of lipid rafts, which are specialized microdomains that organize cell signaling molecules [26]. Interestingly, in the same conditions, different long-chain and very-long-chain ceramides, such as Cer 18:0;O2/16:0, Cer 18:1;O2/16:0, Cer 18:1;O2/22:0, Cer 18:1;O2/24:0, and Cer 18:2;O2/24:0, showed higher levels under SMC compared to the controls. On the other hand, long-chain sphingomyelins SM 16:1;O2/16:0 and SM 16:1;O2/18:1, as well as very-long-chain SM 18:1;O2/24:0 and SM 18:2;O2/24:0, showed lower levels compared to the controls. Interestingly, ceramides were reported to increase the intestinal epithelial cell permeability in a model of Caco-2 cell monolayers [16]. Moreover, Bock et al. were able to obtain naturally occurring long-chain ceramides by adding bacterial sphingomyelinase to Caco-2 cells [27]. Furthermore, the biological chain length-specific properties of ceramides are well reviewed by Grösch et al. [28]. Importantly, in the intestinal tract, ceramides are synthesized through three different pathways: de novo synthesis, salvage, and sphingomyelins hydrolysis (Figure 1). In the first pathway, ceramides are produced in the endoplasmic reticulum with the condensation of palmitoyl-CoA and serine, catalyzed by serine palmitoyl transferase and with enzyme 3-ketosphinganine reductase, ceramide synthase (CerS1-6), and dihydroceramide desaturases (DES1 and DES2) [29]. With the salvage pathway, sphingosine is re-acylated to ceramide. In the intestine, sphingomyelin hydrolysis is catalyzed by neutral sphingomyelinase (SMase), which is expressed in the mucosal cells of the small intestine compared to alkaline sphingomyelinase, which is predominantly found in the jejunum [30]. Sphingomyelin can be catabolized to regenerate ceramides during stress by sphingomyelinases through the sphingomyelin hydrolysis pathway [31].

Finally, we also found an upregulation of phosphatidylethanolamine PE 16:0/16:1, PE 16:1/18:1, and PE 18:0/20:5. High levels of phosphatidylcholine (PC) and phosphatidylethanolamine (PE) are indicative of membrane hyperplasia and have been shown to be correlated with enhanced apoptosis in cancer cell lines subjected to SMC treatment [32,33,34]. 

### 2.2. Caco-2 Cells Treated with DSS

Considering these findings and the role of phospholipids as critical lipid components in intestinal epithelial cells, where they are involved in inflammatory processes [35] and are pivotal in maintaining barrier integrity and nutrient absorption [31], we decided to expose Caco-2 cells to the proinflammatory agent dextran sodium sulfate (DSS) under simulated microgravity conditions. To achieve this, differentiated Caco-2 cells were exposed to a 3% solution of DSS to induce an inflammatory response. After conducting mass spectrometric analysis, data of Caco-2 cells exposed to DSS under terrestrial gravity and simulated microgravity were submitted to OPLS-DA (score plots in Figure 2 and discriminant metabolites in Table 2). The results under SMC indicated a downregulation of long-chain phosphatidylcholines in Caco-2 cells exposed to DSS (Table 2). We also found the long-chain ceramide Cer 18:1;O2/16:0 and the very-long-chain ceramides Cer 18:1;O2/24:0 and Cer 18:1;O2/24:1 to be upregulated (Table 2). Very-long-chain ceramides are synthesized in the endoplasmic reticulum by ceramide synthase 2 (CerS2 or LASS2) and ceramide synthase 5 (CerS5 or LASS5); in particular, Cers2 is able to produce C20:0, C22:0, C24:1, C24:0, C26:1, and C26:0 ceramide, but is unable to synthesize C16:0. Furthermore, CerS 5 may produce C14:0-, C16:0-, C18:0-, and C18:1-ceramides [36,37]. CerS2 and CerS5 are, instead, predominantly expressed in the intestine [38]. Interestingly, in a DSS-induced colitis mouse model, CerS2-deficient and CersS5-deficient mice developed more severe disease symptoms in comparison to wild-type mice [39,40]. Very-long-chain ceramides synthesized by Cers2 are considered lipotoxic and are able to trigger mitochondrial dysfunction, oxidative stress, and cell death in cardiomyocytes [41]. Furthermore, ceramides are dysregulated in Crohn’s disease and ulcerative colitis (UC) patients [42]. Diab et al. reported that very-long-chain ceramides were found to be increased in UC patients [42]. On the contrary, Bazarganipour et al. reported a reduction in de novo sphingolipids synthesis in the colon tissues of UC patients [43].

In the same microgravity experiment, SM 18:1;O2/16:0, SM 18:1;O2/23:0, SM 18:1;O2/24:1, and SM 18:1;O2/26:1 were found to be upregulated (Table 2), contrasting with the results obtained under microgravity but without DSS addition (see Table 1). This upregulation is consistent with the reduced levels of phosphatidylcholines observed in the DSS experiment, given that phosphatidylcholines act as precursors in sphingomyelin synthesis through the sphingomyelin synthase pathway. Furthermore, Qi et al. reported lipidomics evidence in DSS-treated mice, where sphingomyelins (SM) and ceramides were significantly increased in the colitis group compared with the controls [44]. In particular, very-long-chain ceramides disrupt barrier function and aggravates dextran sulfate sodium-induced colitis [40,45], while the long-chain ceramides may contribute to tissue dysfunction [44].

To eliminate the influence of microgravity, we conducted the same experiment with DSS under terrestrial gravity conditions. In this case, we observed higher levels of long- and very-long-chain ceramides, but lower levels of sphingomyelins compared to the controls (Appendix A).

### 2.3. Caco-2 Cells Treated with LPS

We also performed an experiment in which Caco-2 cells were treated with the pro-inflammatory agent LPS at a concentration of 10 µg/mL, as reported in Facchin et al.’s 2022 protocol [46], to investigate lipid alterations under microgravity conditions. By applying an OPLS-DA (score plot in Figure 3), we were able to annotate different discriminating lipids (Table 3) between Caco-2 cells treated with LPS under simulated microgravity conditions vs. controls. In the LPS experiments, long- and very-long-chain phosphatidylcholines were found to be downregulated, while long- and very-long-chain ceramides such as Cer 18:0;O2/16:0, Cer 18:1;O2/16:0, Cer 18:1;O2/24:0, and Cer 18:2;O2/24:0 were found to be downregulated. Furthermore, the glucosylceramide Glc-Cer 18:1;O2/16:0 was found to be downregulated (Table 3). Sphingomyelins such as SM 18:0;O2/16:0, SM 18:1;O2/16:0, SM 18:1;O2/22:0, and SM 18:2;O2/24:0 showed higher levels under SMC, suggesting enhanced enzymatic activity of sphingomyelinase synthase. These findings suggest that, when Caco-2 cells are treated with the proinflammatory agent LPS under microgravity conditions, the de novo synthesis of ceramides is somehow inhibited (Figure 1).

To eliminate the influence of microgravity, we conducted the same experiment with LPS under terrestrial gravity conditions. In this scenario, we observed an upregulation of long- and very-long-chain ceramides compared to controls (Appendix A), which mirrors the results found in the microgravity experiment without treatment (Table 1). These results indicate that microgravity may induce an inflammatory response in Caco-2 cells similar to that triggered by LPS.

### 2.4. mRNA Expression

We also explored the Caco 2 cells’ gene expression of the six genes (*SPTLC1*, *SPTLC2*, *KDSR*, *CERS2*, *CERS6*, and *DEGS1*) involved in the ceramide de novo synthesis using the *TFRC* as an endogenous gene. This analysis revealed that in microgravity conditions (SMC), significantly higher expression of *SPTLC1*, *KDSR*, *CERS6,* and *DEGS1* occurred when Caco2 cells were treated with DSS. When Caco2 cells were treated with LPS, a significant increase in gene expression was measured for *CERS2* and *DEGS1.* On the other hand, when the experiment was replicated under terrestrial gravity conditions, we observed that *SPTLC1*, *SPTLC2*, *KDSR*, *CERS2*, *CERS6,* and *DEGS1* were found to be upregulated when Caco-2 cells were treated with LPS and DSS (Figure 4 and Figure 5).

### 2.5. Inhibition of the Ceramides De Novo Synthesis

To prove the inhibition of de novo ceramide synthesis after Caco-2 cells were treated with LPS, we used myriocin as a selective inhibitor of this pathway. Myriocin is able to inhibit serine palmitoyl-transferase, thus blocking the de novo pathway for ceramide synthesis [47,48]. To further prove this trend, a semiquantitative targeted analysis was conducted to measure the levels of long-chain and very-long-chain Cer, SM, Glc-Cer, and sphingosines (SPH) on Caco-2 cell samples treated with LPS alone and LPS in combination with myriocin. As expected, in the samples treated with myriocin ceramides and glucosylceramides showed lower levels compared to controls, while sphingosines did not change (Figure 6). Additionally, high-resolution mass spectrometric analysis of these samples, reported as relative abundance, showed negligible levels of 3-ketosphinganine, a crucial metabolite in the synthesis of ceramides (Figure 6). In the same experiment, we also measured a lower level of glucosylceramides when cells were treated with myriocin. Taken together, these results confirm the inhibition of ceramide de novo synthesis when caoco-2 cells were treated with LPS and myriocin.

## 3. Materials and Methods

### 3.1. Chemicals

Analytical LC grade methanol, 2-propanol, chloroform, acetonitrile, ammonium acetate and formate, lipopolysaccharide, dextrane sodium suwase, myriocin, and a SPLASH^®^ LIPIDOMIX^®^ deuterated standard component mixture were purchased from Sigma Aldrich (Milan, Italy). Cell culture materials were purchased from Euroclone (Milan, Italy). Bi-distilled water with a conductivity of 0.055 µS/cm was obtained using a MilliQ purification system (Millipore, Milan, Italy).

### 3.2. Microgravity Experiment

Microgravity experiments were conducted using a 3D Random Positioning Machine (RPM) (Fokker Space, Amsterdam, The Netherlands) (Figure 7). The RPM consisted of two perpendicular frames that rotate independently. This setup continuously changed the average value of the gravity vector to zero *g*. The rotation mode and speed were set using specific software connected to the RPM. In this case, the Random Walk mode at 80 degrees/s was selected.

### 3.3. Cell Cultures

Intestinal Caco-2 cells (ECACC Salisbury, Wiltshire, UK) were cultured in Dulbecco’s modified Eagle’s medium (DMEM), supplemented with 10% heat-inactivated bovine serum, 2 mM L-glutamine, 1% non-essential amino acids, 100 U/mL penicillin, and 100 mg/mL streptomycin, in monolayers at 37 °C in a humidified atmosphere of 5% CO_2_ [49], with the medium being replaced twice a week. For experimental studies, Caco-2 cells, at passage 25–40, were plated and used 18–21 post-seeding, when fully differentiated. First, Caco-2 cells were clinorotated for 24 h. Additionally, Caco-2 cells were coincubated with a culture medium containing 3% DSS or LPS (10 μg/mL) for 24 h, respectively. The cells were grown inside 75 cm^2^ plug seal screw-cap flasks, and each flask was filled with culture medium before being placed inside the clinostat. Both the control 1 *g* samples and those subjected to SMC were collected at 24 h. After the clinorotation, the culture medium was removed, and the cellular material was recovered by adding 2 mL of trypsin to detach cells and, subsequently, 2 mL of culture medium to inactivate trypsin digestion. Subsequently, the cellular material was taken from the flasks, and after centrifugation, the cell pellet was stored at −80 °C until the lipidomics analysis. In the inhibition experiment targeting de novo ceramide synthesis, Caco-2 cells were co-cultured in Petri dishes with myriocin for 48 h at 1 *g*. After 24 h, LPS (10 μg/mL) was introduced, and samples were collected at 24 h.

### 3.4. mRNA Expression Analysis

RNA was extracted from Caco-2 cells using the RNAeasy Mini Kit (Qiagen, Hilden, Germany), following the manufacturer’s protocol. An aliquot of 2 μg RNA/sample was retrotranscribed using the High-Capacity Kit (Applied Biosystems, Carlsbad, CA, USA). Gene expression was evaluated on a QuantStudio™ 5 Real-Time PCR System (Thermo Fisher Scientific, Waltham, MA, USA) by qPCR using PowerUp SYBR Green Master Mix (Applied Biosystems, Carlsbad, CA, USA) for each gene tested, SPTLC1, SPTLC2, KDSR, CERS2, CERS6, DEGS1, and the endogenous gene TFRC. Primer sequences can be found in Appendix A. PCR conditions were primary denaturation at 95 °C for 2 min followed by 50 cycles of denaturation at 95 °C for 15 s and annealing/extension at 60 °C for 1 min. After the amplification cycles, melting curves were produced by increasing the temperature from 65 °C to 95 °C, holding each temperature for 5 s, and reading the fluorescence every 0.5 °C. Statistical analyses were performed by comparing the average ΔCt of the sample groups of interest using Student’s *t*-test (Appendix A).

### 3.5. Sample Preparation

Cells were extracted following the Folch extraction procedure with a slight modification [50]. In detail, 3 mg of cells was extracted using 375 μL of a methanol and chloroform mixture (2:1, *v*/*v*). Samples were vortexed every 15 min for up to 1 h, when 350 μL of chloroform and 90 μL of KCl solution 0.2 M were subsequently added. The obtained solution was centrifuged at 17,700 relative centrifugal force for 10 min, and 400 μL of the organic layer was transferred into a glass vial and dried under a gentle nitrogen stream. The dried chloroform phase was reconstituted with 20 μL of a methanol and chloroform mixture (1:1, *v/v*) and 80 μL 2-propanol: acetonitrile:water mixture (2:1:1, *v*/*v*).

### 3.6. Untargeted UHPLC-QTOF/MS Analysis

Cell samples were analyzed in triplicate using an Agilent UHPLC-QTOF/MS 6560 coupled with a 1290 Infinity II LC system (Agilent Technologies, Palo Alto, CA, USA) injecting 1 μL in the ESI positive ionization mode and 4 μL in the negative ionization mode, respectively. Chromatographic separation was achieved with a C18 Kinetex column (1.5 μm, 100 mm × 2.1 mm, Phenomenex, Bologna, Italy) kept at 50 °C with a flow rate of 0.4 mL/min.

The mobile phase for the positive ionization mode was: (A) 10 mM ammonium formate in 60% Milli-Q water and 40% acetonitrile, and (B) 10 mM ammonium formate in 90% 2-propanol and 10% acetonitrile. The solvent gradient employed began with 80% A, decreasing linearly to 50% A over 2.1 min, then to 30% A over the subsequent 10 min. This was followed by a further reduction to 1% A for 1.9 min before returning to the initial conditions within 1 min. For negative ionization mode, 10 mM ammonium acetate was used in place of ammonium formate.

An Agilent jetstream source was operated using a gas temperature of 200 °C, nitrogen gas flow of 10 L/min, nebulizer gas (nitrogen) of 50 psig, sheath gas temperature of 300 °C, sheath gas flow of 12 L/min, capillary voltage of 3500 V for positive mode and 3000 V for negative mode, nozzle voltage of 0 V, fragmentor of 150 V, scan mass range of 50–1700 *m*/*z*, and collision energy of 20 eV for positive ionization mode and 25 eV for negative ionization mode prior to analysis. Data acquisition was carried out using the Agilent MassHunter LC/MS Acquisition console (Agilent Technologies, Palo Alto, CA, USA). Quality control (QC) samples were prepared by pooling an aliquot of 10 μL from each sample. Samples and QCs were injected into the UHPLC-QTOF/MS system and acquired in the positive and negative ionization modes.

### 3.7. Lipid Annotation

Data were also acquired using the auto MS/MS acquisition mode to annotate complex lipids and obtain fragmentation spectra. This technique operated in iterative mode with a mass error tolerance of 10 ppm. Samples were injected 5 times in positive and negative mode with 1 and 4 μL per injection, respectively. The data were processed using Agilent Lipid Annotator software, version 1.0, (Agilent Technologies, Santa Clara, CA, USA).

### 3.8. Targeted UHPLC-MS/MS Analysis

Chromatographic separation of ceramides, sphingomyelins, glucosylceramides, and sphingosines was performed using a 1260 Agilent infinity II multisampler BIO binary pump (Agilent Technologies, Palo Alto, CA, USA). The chromatographic system was connected to an Agilent Ultivo triple quadrupole mass spectrometer (Agilent Technologies, Santa Clara, CA, USA). A Kinetex C18, 1.5 μm, 100 mm × 2.1 mm chromatographic column (Phenomenex, Bologna, Italy) was used. The mobile phase consisted of water (A) and acetonitrile (B) (10:90, *v*/*v*) containing 0.1% of formic acid with a flow rate of 0.3 mL/min. The gradient was 99% (A) as the initial condition, then 0–1 min linear to 80% (A), kept for 7 min, and linear to 10% (A) from 8 to 12 min, followed by 2 min at the initial conditions for re-equilibration

Mass spectrometric analysis was performed in the positive ionization mode. Nitrogen was used as the sheath gas, drying gas, and collision gas. The sheath gas and drying gas flow rates were set at 11 L/min at 375 °C and 7 L/min at 300 °C, respectively. The nebulizer was set at 45 psi and the capillary voltage and nozzle voltage were set at 4000 V and 1500 V, respectively. The fragmentor and cell acceleration voltage were 135 V and 9 V, respectively. The acquisition was performed in the multiple reaction monitoring (MRM) mode. The MS system was controlled by Agilent Mass Hunter Workstation Data Acquisition. Mass characteristics of sphingolipids are reported in Appendix A.

### 3.9. Data Analysis

Data were pre-processed using the Mass Profinder 10.0 software (Agilent Technologies, Santa Clara, CA, USA), which enables mass recalibration and deconvolution, providing a data matrix that includes all features present across all samples. Mass spectrometry features were filtered according to their presence in QC samples, with a threshold set at 20%. The resulting data matrix was processed using SIMCA 15.0 software (Sartorius, Umeå, Sweden). Initially, data were autoscaled by UV scaling, and then a Principal Component Analysis (PCA) was performed (Appendix A). This unsupervised method aids in visualizing the distribution of samples and variables in a multivariate space, as well as in highlighting deviating features and the presence of outliers. Subsequently, a partial Least Squares Discriminant Analysis (PLS-DA) and its orthogonal extension (OPLS-DA) were employed. These supervised classification models were used to visualize and evaluate the discriminative variables between samples assigned to different classes. The variable importance in the models is reported as VIP score, and values >  1.0 were deemed as discriminant. The outcomes of the OPLS-DA analyses were validated by testing 400 permutations (Appendix A) [51]. The statistical significance of the mean difference was also assessed using Student’s *t*-test.

## 4. Conclusions

This study provides critical insights into the impact of microgravity on the lipid metabolism of Caco-2 cells, specifically highlighting the alterations in complex lipid profiles under simulated microgravity conditions (Appendix A). The results demonstrate that microgravity induces significant remodeling of cellular membranes, as evidenced by the upregulation of various PCs and LPCs, alongside an increase in the levels of long-chain and very-long-chain ceramides. These findings suggest that microgravity enhances the synthesis of specific lipid species that play crucial roles in maintaining membrane integrity, fluidity, and signaling pathways, which are vital for the barrier function of the intestinal epithelium. Furthermore, the study observed differential lipid responses when Caco-2 cells were exposed to pro-inflammatory agents such as DSS and LPS under microgravity. The combination of microgravity and inflammatory agents intensified the dysregulation of lipid metabolism, particularly the downregulation of phosphatidylcholines and the upregulation of ceramides and sphingomyelins. These findings were validated through the relative quantification of mRNA gene expression related to de novo ceramide synthesis. Notably, in SMC conditions, the upregulation of *SPTLC1*, *KDSR*, *CERS6*, and *DEGS1* was observed in cells treated with DSS. Specifically, *CERS6* is involved in the synthesis of C16:0 ceramides, which were found to be upregulated in the lipidomics analysis. Conversely, in the LPS experiment, we observed an upregulation of *SPTLC1*, *KDSR*, *CERS2*, and *DEGS1*. In this context, *CERS2* is responsible for producing very-long-chain fatty acid ceramides (C20–C26), consistent with the lipidomics findings that showed an increase in C24:0 ceramides. Taken together, these results suggest that microgravity may amplify inflammatory responses at the cellular level, which could have implications for gastrointestinal health in spaceflight conditions. Moreover, the inhibition of de novo ceramide synthesis through myriocin confirmed the role of this pathway in the observed lipid alterations. The reduction in ceramide levels following the inhibition of serine palmitoyl-transferase underlines the importance of de novo synthesis in ceramide accumulation during microgravity and inflammatory conditions.

In conclusion, the findings of this study emphasize the profound effects of microgravity on lipid metabolism in intestinal epithelial cells, with potential implications for the development of strategies to mitigate the adverse effects of long-term space travel on the gastrointestinal system.

While this study provides valuable insights, it has several limitations. The use of Caco-2 cells, despite being a widely accepted model, may not fully replicate the complexity of the intestinal epithelium. Additionally, the SMC, although informative, is not identical to the true microgravity experienced during spaceflight. Future research should involve in vivo studies and spaceflight experiments to validate these findings.

## Figures and Tables

**Figure 1 ijms-25-12638-f001:**
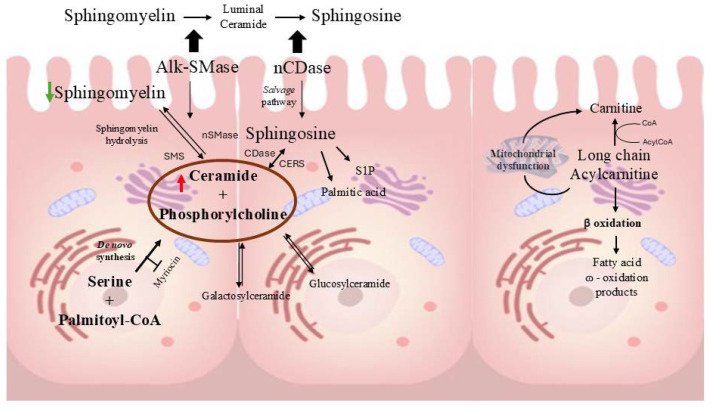
Ceramide pathway within the intestinal lumen, highlighting the key steps involved in ceramide metabolism and fatty acid oxidation. S1P: sphingosine-1-phosphate, SMS: sphingomyelinsynthases, AlkSMase: alkaline sphingomyelinase, nCDase: neutral ceramidase. The red narrow indicates an increase in ceramides, while the green narrow shows a decrease in sphingomyelins.

**Figure 2 ijms-25-12638-f002:**
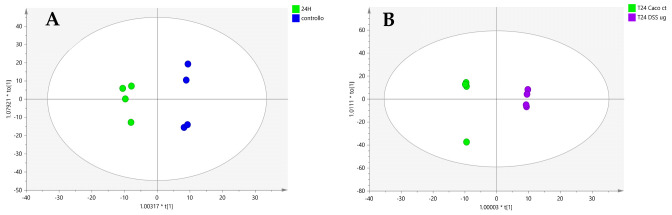
OPLS-DA score plots of Caco-2 cells treated with DSS under terrestrial gravity (**A**) and Caco-2 cells treated with DSS under simulated microgravity (**B**). In plot A, control samples are represented by blue circles, while green circles represent samples treated with DSS. In plot B, control samples are represented with green circles, while violet circles represent samples treated with DSS under simulated microgravity. Validation parameters for the statistical model are indicated with R^2^Y and Q^2^ values.

**Figure 3 ijms-25-12638-f003:**
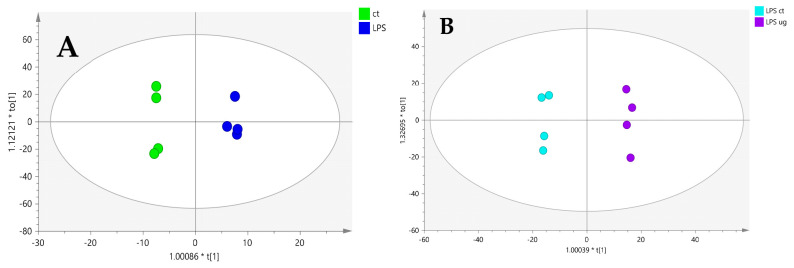
OPLS-DA score plots of Caco-2 cells treated with LPS under terrestrial gravity (**A**) and Caco-2 cells treated with LPS under simulated microgravity (**B**). In plot A, control samples are represented by green circles, while blue circles represent samples treated with LPS. In plot B, control samples are represented with light blue circles, while violet circles represent samples treated with LPS under simulated microgravity. Validation parameters for the statistical model are indicated with R^2^Y and Q^2^ values.

**Figure 4 ijms-25-12638-f004:**
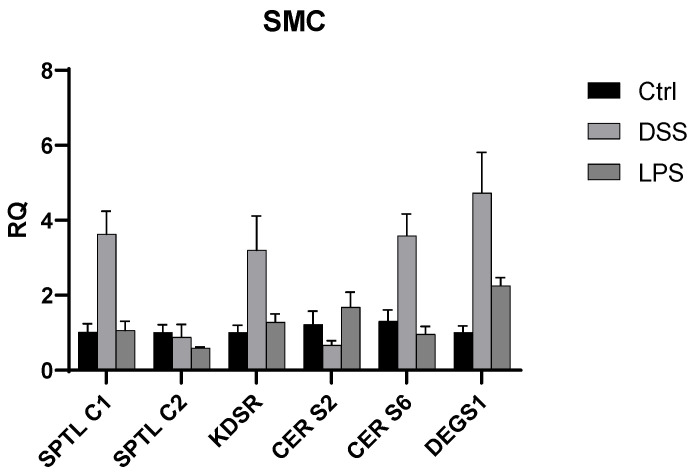
The bar graph compares the relative quantification (RQ) of various gene expressions (*SPTLC1*, *SPTLC2*, *KDSR*, *CERS2*, *CERS6*, and DEGS1) in SMC (0 *g*) under three conditions: Ctrl g0 (black), DSS g0 (light gray), and LPS g0 (dark gray). The results suggest an increased gene expression in the DSS g0 and LPS g0 conditions compared to the control, with *DEGS1* showing the higher expression level, especially under the DSS g0 condition. Error bars indicate the standard deviation of gene expression levels across the samples.

**Figure 5 ijms-25-12638-f005:**
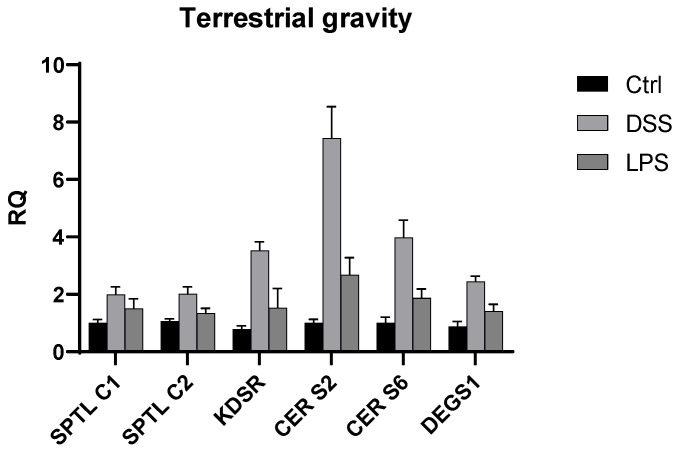
The bar graph of the relative quantification (RQ) of various gene expressions (*SPTLC1*, *SPTLC2*, *KDSR*, *CERS2*, *CERS6*, and *DEGS1*) in terrestrial gravity conditions (1 *g*) for three groups: Ctrl g1 (black), DSS g1 (light gray), and LPS g1 (dark gray). Gene expression is generally higher in the DSS g1 and LPS g1 conditions compared to the control. Notably, *CER S2* showed the highest increase in expression, particularly in the LPS g1 and DSS g1 groups. Error bars indicate the standard deviation of gene expression levels across the samples.

**Figure 6 ijms-25-12638-f006:**
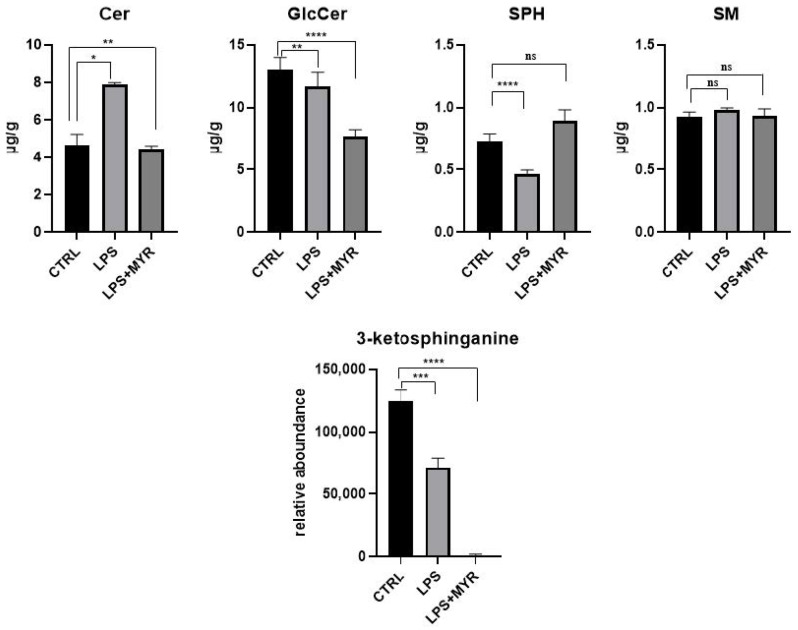
Levels of ceramides, glucosylceramides, and sphingosines in Caco-2 cells expressed in µmol/g: comparison among control samples, LPS, and LPS with myriocin treatment. **** *p* < 0.00001, *** *p* < 0.0001, ** *p* < 0.001, * *p* < 0.01, ns *p* > 0.05. Relative abundance of the lipid 3-ketosphinganine in LPS and LPS + myriocin (MYR).

**Figure 7 ijms-25-12638-f007:**
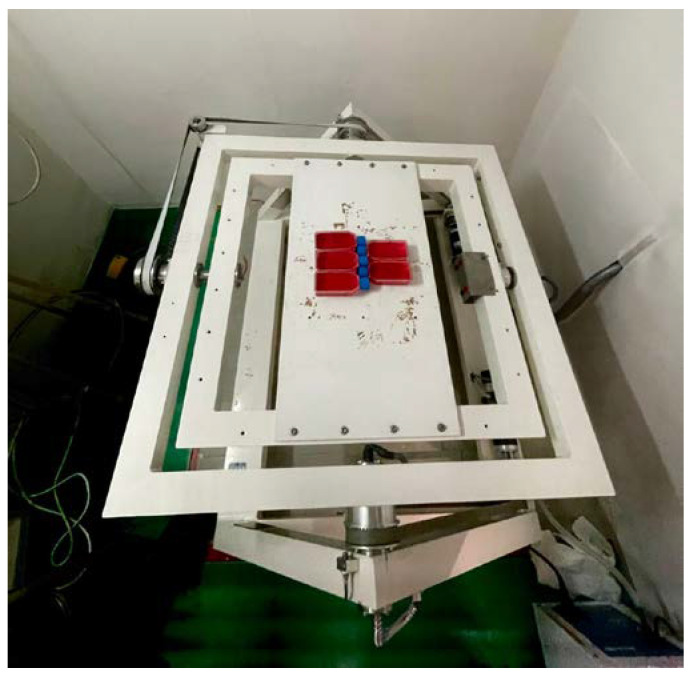
Three-dimensional rotating clinostat used to simulate microgravity (University of Sassari, Italy).

**Table 1 ijms-25-12638-t001:** OPLS-DA discriminant lipids of Caco-2 cells subjected to simulated microgravity conditions (SMC) for 24 h. Cer: ceramide, LPC: lysophosphatidylcholine, PC: phosphocholine, PE: phosphatidylethanolamine, SM: sphingomyelin, * *p* < 0.05; ** *p* < 0.01; *** *p* < 0.001. Δ ppm: difference between the theoretical mass and the measured mass expressed as part per million. Mean differences were tested for statistical significance using Student’s *t*-test (n = 3). The signs “+” and “-” indicate upregulated and downregulated lipids in SMC, respectively.

Attribution	Theor. *(m*/*z)*	Exp. *(m*/*z)*	Δ ppm	Rt (min)	*p* Value	Regulation in SMC
Cer 18:0;O2/16:0	540.5350	540.5348	−0.51	4.59	**	+
Cer 18:1;O2/16:0	538.5194	538.5198	−0.52	4.24	**	+
Cer 18:1;O2/22:0	622.6133	622.6130	−1.05	7.68	***	+
Cer 18:1;O2/24:0	650.6446	650.6441	−0.43	8.12	***	+
Cer 18:2;O2/24:0	648.6289	648.6286	−0.28	7.84	***	+
LPC 16:0	496.3398	496.3392	−0.74	0.94	*	+
LPC 18:0	524.3711	524.3710	−1.05	1.12	*	+
LPC 18:1	522.3554	522.3550	−0.88	0.96	*	+
PC 16:0/16:1	732.5538	732.5537	−0.13	3.20	**	+
PC 16:1/16:1	730.5381	730.5385	0.54	2.65	**	+
PC 16:1/18:0	760.5851	760.5853	−0.13	4.47	**	+
PC 16:1/18:1	758.5694	758.5692	−0.39	3.80	**	+
PC 15:0/20:4	768.5538	768.5531	−0.10	4.05	**	+
PC 18:0/18:2	786.6007	786.6002	−0.63	5.03	**	+
PC 18:0/18:3	784.5851	784.5858	−0.52	4.86	**	+
PC 18:0/20:1	816.6477	816.6476	−0.48	6.68	**	+
PE 16:0/16:1	690.5068	690.5069	−0.51	3.27	**	+
PE 16:1/18:1	716.5244	716.5240	−0.30	3.30	**	+
PE 18:0/20:5	766.5381	766.5430	6.39	3.25	*	+
SM 16:1;O2/16:0	675.5435	675.5432	−0.44	2.71	**	-
SM 16:1;O2/18:1	701.5592	701.5598	−0.28	2.82	***	-
SM 18:1;O2/24:0	815.7000	815.7001	−0.85	7.82	***	-
SM 18:2;O2/24:0	813.6844	813.6843	−0.49	6.46	***	-

**Table 2 ijms-25-12638-t002:** OPLS-DA discriminant lipids of Caco-2 cells subjected to SMC and treated with DSS compared to controls. SMC for 24 h. Cer: ceramide, PC: phosphocholine, SM: sphingomyelin. Δ ppm: difference between the theoretical mass and the measured mass expressed as part per million. The signs “+” and “-” indicate upregulated and downregulated lipids in SMC, respectively.

Attribution	Theor. (*m*/*z*)	Exp. (*m*/*z*)	Δ ppm	Rt (min)	VIP	Regulation in SMC
Cer 18:1;O2/16:0	538.5194	538.5191	−0.52	4.24	1.01	+
Cer 18:0;O2/22:0	624.6289	624.6290	172	5.51	1.02	+
Cer 18:1;O2/24:0	650.6446	650.6441	−0.43	8.12	1.25	+
Cer 18:2;O2/24:0	648.6289	648.6285	−0.46	7.84	1.01	+
PC 14:0/16:1	704.5225	704.5220	0.14	2.23	1.22	-
PC 15:0/16:1	718.5381	718.5387	0.41	2.47	1.53	-
PC 16:0/16:1	732.5538	732.5536	1.91	3.20	1.10	-
PC 16:1/16:1	730.5381	730.5383	1.36	2.65	1.13	-
PC 18:0/18:1	788.6164	788.6163	1.77	5.65	1.20	-
PC 17:1/21:0	816.6477	816.6471	0.01	4.57	1.15	-
SM 18:1;O2/16:0	703.5748	703.5744	1.70	3.43	1.73	+
SM 18:1;O2/23:0	801.6844	801.6845	−5.29	7.35	1.57	+
SM 18:1;O2/24:1	813.6844	813.6840	−0.49	6.46	1.16	+
SM 18:1;O2/26:1	841.7157	841.7156	−0.11	8.10	1.15	+

**Table 3 ijms-25-12638-t003:** OPLS-DA discriminant lipids of Caco-2 cells treated with LPS and subjected to SMC for 24 h. Cer: ceramide, Glc-Cer: glucosylceramide, LPC: lysophosphatidylcholine, PC: phosphocholine, PE: phosphatidylethanolamine, PS: phosphatidylserine, SM: sphingomyelin. Cer: ceramide, PC: phosphocholine, SM: sphingomyelin. Δ ppm: difference between the theoretical mass and the measured mass expressed as part per million. The signs “+” and “-” indicate upregulated and downregulated lipids in SMC, respectively.

Attribution	Theor. *(m*/*z)*	Exp, *(m*/*z)*	Δ ppm	Rt (min)	VIP	Regulation in SMC
Cer 18:0;O2/16:0	538.5205	538.5201	0.55	4.59	1.10	-
Cer 18:1;O2/16:0	538.5194	538.5198	−0.52	4.24	1.01	-
Cer 18:1;O2/24:0	650.6446	650.6441	−0.94	8.12	1.03	-
Cer 18:2;O2/24:0	648.6289	648.6287	−0.77	7.84	1.02	-
Glc-Cer 18:1;O2/16:0	700.5722	700.5725	0.42	3.90	1.12	-
LPC 16:0	496.3398	496.3397	0.60	0.94	1.47	+
PC 15:1/16:0	718.5381	718.5384	0.55	2.70	1.28	-
PC 15:1/16:1	716.5225	716.5222	0.69	2.50	1.50	-
PC 18:1/14:0	732.5538	732.5534	−0.27	4.82	1.13	-
PC 15:1/18:0	746.5694	746.5693	0.40	4.24	1.15	-
PC 18:0/16:0	762.6007	762.6001	−0.39	6.01	1.13	-
PC 18:1/18:0	788.6164	788.6162	0.63	5.76	1.62	-
PC 18:1/18:1	786.6007	786.6001	−0.38	4.77	1.34	-
PC 18:1/20:1	814.6320	814.6329	0.73	5.68	1.33	-
PE 18:1/18:0	746.5694	746.5692	−0.13	4.24	1.42	-
PE 18:1/20:0	774.6007	774.6003	0.51	5.19	1.34	-
PE 18:2/20:0	772.5851	772.5850	−0.25	4.33	1.10	-
PS 16:0/17:0	750.5280	750.5285	0.79	5.99	1.26	+
SM 18:0;O2/16:0	705.5905	705.5900	−0.56	3.60	1.21	+
SM 18:1;O2/16:0	703.5748	703.5743	0.42	3.43	1.53	+
SM 18:1;O2/22:0	787.6687	787.6681	−0.12	6.32	1.18	+
SM 16:1;O2/24:0	815.7000	815.7002	0.36	6.08	1.18	+
SM 18:2;O2/24:0	813.6844	813.6840	−0.73	6.46	1.10	+

## Data Availability

The data presented in this study are available on request from the corresponding author.

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
