# Peer review of "Lipidomics of Caco-2 Cells Under Simulated Microgravity Conditions"

_ijms, 2024, doi:10.3390/ijms252312638_

Round 1

Reviewer 1 Report

Comments and Suggestions for Authors

Although the article presents several interesting aspects, a number of inaccuracies and omissions significantly diminish its overall value.

First, the authors should align the nomenclature system with the guidelines established by Liebish et al. (10.1194/jlr.S120001025 and 10.1194/jlr.M033506) Specifically, for sphingolipids, the number of hydroxylations should be clearly indicated, as is standard practice.

In the materials and methods section, the authors apply a 40% threshold for variable selection, which is unusually high—typically, it does not exceed 20% (10.1038/nprot.2011.335). What is the rationale behind using such an elevated threshold?

Additionally, the authors reference the use of QC in the materials and methods section. To provide a clearer understanding of the analytical variability, PCA plots with QC should be included, possibly as supplementary material (10.1038/nprot.2011.335). Moreover, the order of sample injection and extraction should be explicitly stated, as variability observed in the results could be attributed to these procedural factors.

Furthermore, the authors mention the use of deuterated standards in the materials and methods section, but these are not referenced in the results or discussion. Why is this detail missing?

The explanation of mobile phase B composition for targeted analyses is also unclear. The use of two different gradients for targeted and untargeted analyses is puzzling, particularly as these approaches typically yield different types of information. Since no specific extraction method or acidic hydrolysis of phospholipids was employed, the rationale for this approach is questionable, potentially resulting in the loss of accurate mass information, which is crucial for proper identification. Additionally, the use of different additives for positive and negative mode analyses may introduce discrepancies in retention mechanisms. 

It appears, moreover, that insufficient attention was paid to the preparation of the final manuscript. The citation style is inconsistent—authors' names are used in the text, but numbered references are listed in the bibliography.

Figure 5 appears to be a screenshot of an image. I would recommend replacing it with a high-quality version. While these errors do not directly affect the scientific content of the paper, they detract from the overall professional presentation.

I strongly urge the authors to correct these errors and inconsistencies and take greater care in preparing the next submission. Unfortunately, I must reject the paper in its current form, but I encourage resubmission after careful adjustments in line with the requested revisions.

Author Response

Reviewer 1

Although the article presents several interesting aspects, a number of inaccuracies and omissions significantly diminish its overall value.

First, the authors should align the nomenclature system with the guidelines established by Liebish et al. (10.1194/jlr.S120001025 and 10.1194/jlr.M033506) Specifically, for sphingolipids, the number of hydroxylations should be clearly indicated, as is standard practice.

Authors: we changed the lipid nomenclature as requested by the reviewer and following the Liebish et al. manuscript.

In the materials and methods section, the authors apply a 40% threshold for variable selection, which is unusually high—typically, it does not exceed 20% (10.1038/nprot.2011.335). What is the rationale behind using such an elevated threshold?

Authors: We usually use a 20% threshold variable selection, in this case we used a 40% threshold value because of low number of monolayer cells available and the reduced MS abundance.

Additionally, the authors reference the use of QC in the materials and methods section. To provide a clearer understanding of the analytical variability, PCA plots with QC should be included, possibly as supplementary material (10.1038/nprot.2011.335). Moreover, the order of sample injection and extraction should be explicitly stated, as variability observed in the results could be attributed to these procedural factors.

Authors: we provide a new figure in the supplementary materials section. QCs were injected first for 10 times then SPLASH std was injected at two levels (high and low) for three times. Afterward, real samples were injected randomly.

Furthermore, the authors mention the use of deuterated standards in the materials and methods section, but these are not referenced in the results or discussion. Why is this detail missing?

Authors: we used a splash lipid deuterated standard mixture. We reported it in the MM section as requested. 

The explanation of mobile phase B composition for targeted analyses is also unclear. The use of two different gradients for targeted and untargeted analyses is puzzling, particularly as these approaches typically yield different types of information. Since no specific extraction method or acidic hydrolysis of phospholipids was employed, the rationale for this approach is questionable, potentially resulting in the loss of accurate mass information, which is crucial for proper identification. Additionally, the use of different additives for positive and negative mode analyses may introduce discrepancies in retention mechanisms. 

Authors: we used two different MS platforms. An ion mobility QTOF MS for the untargeted analysis and a LCMSMS for the targeted analysis both use different mobile phases mainly to reduce run times.  In our case, additives for positive and negative ionization mode were used to improve lipids signals.  We used the Folch extraction method which is widely used in untargeted analysis. We are planning to make in the future new experiments trying to improve the extraction of ceramides/sphingomyelins. 

It appears, moreover, that insufficient attention was paid to the preparation of the final manuscript. The citation style is inconsistent—authors' names are used in the text, but numbered references are listed in the bibliography.

Authors: we apologize for the inconsistency of the references and style. We changed it accordingly.

Figure 5 appears to be a screenshot of an image. I would recommend replacing it with a high-quality version. While these errors do not directly affect the scientific content of the paper, they detract from the overall professional presentation.

Authors: we changed Figure 5 now Figure 7 adding a high quality version.

I strongly urge the authors to correct these errors and inconsistencies and take greater care in preparing the next submission. Unfortunately, I must reject the paper in its current form, but I encourage resubmission after careful adjustments in line with the requested revisions.

Authors: we thank the reviewer for the encouraging words.

Reviewer 2 Report

Comments and Suggestions for Authors

In this manuscript the authors show the result of an experiment evaluating the impact of microgravity on the lipidome of human cell line in both normal and proinflammatory conditions. Significant differences are reported in the abundance of several glycerophospholipids and ceramides. The experiments are clearly designed and the analyses provide interesting information.

Comments:

-            Table1: I am not sure that “sum composition” is a clear term to define the contents of the second column; I think “hydrophobic chain composition" would be more precise.

-            Line 203: the LPC that are listed are not all hydrolysis products of the PC mentioned on the previous line. Probably only LPC(16:0).

-            It should be mentioned how many different lipid species are detected and quantified in total.

-            Line 226: sphingomyelins are not ceramide hydrolysis products. The results suggest a higher conversion of sphingomyelins into ceramides.

-            Line 246: not clear what the authors mean by “membrane proliferation”. I guess what they mean, but it is not a correct term in my opinion.

-            On page 7 and figure 2 there is a nice description of the pathways leading to ceramide production, but there is no evoked hypothesis from the results obtained. Which pathway is upregulated or downregulated? Why? And what could be done to confirm the point?

-            The way the results are presented, it seems that it is ceramide changes that prompt the test by DSS. However, the changes observed in phospholipids also.

-            The decrease in PC species observed in the presence of DSS is interesting in that it suggests an activation of phospholipase A2 activity. It would be interesting to check whether the expression of phospholipases A2 is increased.

-            In the same way, the expression of, at least, Ceramide synthase 2, should be analyzed. The manuscript seems to insist on the importance of ceramides.  A number of key genes involved in the ceramide pathways should be tested for expression, at least at the RNA level.

-            Sphingomyelin results should be shown on Figure 5, even if not changed.

-            The results of 3-ketosphinganine levels after LPS challenge should be explained, interestingly they show the same pattern as glucosylceramide.

-            Figure 5 would be more eloquent if the results of LPS in normal gravity and in microgravity conditions are shown in parallel.

-            A final cartoon synthesizing the major results in the different experimental conditions would be helpful to appreciate the whole picture.

Comments on the Quality of English Language

The English is basically fine

Author Response

Reviewer 2

In this manuscript the authors show the result of an experiment evaluating the impact of microgravity on the lipidome of human cell line in both normal and proinflammatory conditions. Significant differences are reported in the abundance of several glycerophospholipids and ceramides. The experiments are clearly designed and the analyses provide interesting information.

Comments:

-            Table1: I am not sure that “sum composition” is a clear term to define the contents of the second column; I think “hydrophobic chain composition" would be more precise.

Authors: We changed the definition of sum composition as requested by the reviewer.

-            Line 203: the LPC that are listed are not all hydrolysis products of the PC mentioned on the previous line. Probably only LPC(16:0).

Authors: We do not agree with the reviewer, PC(18:0;18:2) and PC(16:1;18:1) may hydrolize to LPC(18:0) and LPC(18:1).

-            It should be mentioned how many different lipid species are detected and quantified in total.

Authors: we added this information to the Results and discussion section. 

-            Line 226: sphingomyelins are not ceramide hydrolysis products. The results suggest a higher conversion of sphingomyelins into ceramides.

Authors: we agree with the reviewer suggestion. We deleted the part referred to the hydrolysis products.

-            Line 246: not clear what the authors mean by “membrane proliferation”. I guess what they mean, but it is not a correct term in my opinion.

Authors: we agree with the reviewer, we changed proliferation with hyperplasia.

-            On page 7 and figure 2 there is a nice description of the pathways leading to ceramide production, but there is no evoked hypothesis from the results obtained. Which pathway is upregulated or downregulated? Why? And what could be done to confirm the point?

Authors: we changed Figure 2 adding the upregulation of ceramide pathway ans SM pathway downregulation.

-            The way the results are presented, it seems that it is ceramide changes that prompt the test by DSS. However, the changes observed in phospholipids also.

Authors: We changed the the first sentence of 3.2 section.

-           The decrease in PC species observed in the presence of DSS is interesting in that it suggests an activation of phospholipase A2 activity. It would be interesting to check whether the expression of phospholipases A2 is increased.

Authors: unfortunately, we did not measure the phospholipase A2 activity in our experiment. We are planning another experiment to assay different phospholipase activities.

-            In the same way, the expression of, at least, Ceramide synthase 2, should be analyzed. The manuscript seems to insist on the importance of ceramides.  A number of key genes involved in the ceramide pathways should be tested for expression, at least at the RNA level.

Authors: We thank again the reviewer. In this case we were able to measure the relative quantitation of gene expression related to the de novo ceramide pathways. In particular. SPTLC1-2, KDSR, CERS2-6 and DEGS1 genes regulations were measured on Caco2 cells. 

-            Sphingomyelin results should be shown on Figure 5, even if not changed.

Authors: we added the pattern of sphingomyelins to Figure 5

-            The results of 3-ketosphinganine levels after LPS challenge should be explained, interestingly they show the same pattern as glucosylceramide.

Authors: we changesd the section 3.6 adding the sentence “In the same experiment we also measured a lower level of glucosylceramides when cells were treated with myriocin.”

-            Figure 5 would be more eloquent if the results of LPS in normal gravity and in microgravity conditions are shown in parallel.

Authors: we reported in the supplementary section the results of DSS and LPS experiments in terrestrial gravity.

-            A final cartoon synthesizing the major results in the different experimental conditions would be helpful to appreciate the whole picture.

Authors: we added a cartoon recapitulating the experimental data in the supplementary section.

Reviewer 3 Report

Comments and Suggestions for Authors

Your study provides valuable insights into the impact of simulated microgravity on lipid metabolism in Caco-2 cells, particularly focusing on ceramides and sphingomyelins. Below are my comments and questions:

 Questions and Comments

1. Table 1: What is the definition of regulation in SMC? The definition affects the pathological effect of this study. Therefore, a detailed description is needed.

2. Figure 2: The pathway might be related to Table 1. The current figure is hard to understand what molecules changed in this study. Please consider revising the figure for clarity.

3. Tables 1-3: I wondered what molecules changed in common or differed in each environment (microgravity or inflammation model). I recommend a summary of the changing results for better comparison.

Author Response

Reviewer 3

Your study provides valuable insights into the impact of simulated microgravity on lipid metabolism in Caco-2 cells, particularly focusing on ceramides and sphingomyelins. Below are my comments and questions:

 Questions and Comments

  1. Table 1: What is the definition of regulation in SMC? The definition affects the pathological effect of this study. Therefore, a detailed description is needed.

Authors: we changed the results and discussion section accordingly.

  1. Figure 2: The pathway might be related to Table 1. The current figure is hard to understand what molecules changed in this study. Please consider revising the figure for clarity.

Authors: we changed Figure 2 adding the up or down regulation of lipid species.

  1. Tables 1-3: I wondered what molecules changed in common or differed in each environment (microgravity or inflammation model). I recommend a summary of the changing results for better comparison.

Authors: we thank thereviewer for raising this point. We added a cartoon to better explain and compare our results.

Round 2

Reviewer 1 Report

Comments and Suggestions for Authors

While I find the exploration of microgravity intriguing, unfortunately, the manuscript contains experimental errors and issues in the interpretation of lipidomic data that prevent me from accepting the work.

My main concern derives from the answer to one of my previous questions.

The score plot in Figure S1 shows a QC sample entirely separated from the others, raising concerns. With such a score plot, it seems that analytical variability exceeds biological variability. If valid reasons exist for excluding that QC sample, the data should be re-evaluated, potentially lowering the acceptance criterion for variables from 40 to 20%. Removing this QC, which hopefully is due to the first chromatographic run, a good QC clustering has been obtained, and thus it is unclear why the 40% threshold has selected.

Could the authors clarify if data were autoscaled prior to PCA?

According to the authors’ responses, it appears that QC samples were only injected at the start of the run to condition the system, rather than throughout the analysis. If so, this approach does not align with standard metabolomics practices. Indeed, QCs should be injected between samples analyses to ensure the stability of the system.

I must reiterate that an acceptance criterion this broad is not suitable for robust analysis.

Table 1 contains multiple nomenclature inconsistencies. For instance, the symbol ";" is used to separate chains in PE and SM, which is unconventional; either "/" or "_" should be used depending on whether regiochemistry is known. The nomenclature for sphingomyelins follows the older system, while ceramides use the more current one. A consistent approach is necessary.

Additionally, some chromatographic assignments appear unusual. For example, PC 18:0/18:2;O elutes at 4.01 minutes, while PC 18:0/18:3;O elutes at 5.39 minutes. In general, increasing the number of double bonds should decrease retention time in RP chromatography. Are the authors confident in these assignments? I would expect that other, potentially more abundant lipids were also identified, and comparing these results with existing data could be informative.

A detailed review of data and terminology is needed, as the nomenclature in the table differs from that in the text. For PC lipids, the acyl chains are separated by ";" in the text, which appears to correspond to "_" in the authors' nomenclature.

In the table, for PC, the second acyl chain is consistently hydroxylated, and regiochemistry is inferred.

This goes beyond a simple nomenclature error. Although readers may infer from m/z values that PCs are not hydroxylated, the current wording could mislead non-experts familiar with the nomenclature system. They might interpret it as suggesting that hydroxylation occurs selectively during microgravity experiments, consistently on the sn-2 chain, potentially implying stereoselective processes. Therefore, careful attention to this aspect is essential!

Some of the MS/MS data provided in the supplementary material may not be sufficiently informative. For instance, the water loss observed in certain Cer does not offer additional insights; it might be more beneficial to include the signal for the sphingoid base as you have done for other cases, even if this is not likely the base peak. Some assignments also seem potentially incorrect. For example, in positive ion mode, SM typically form m/z 184 as the base peak. You have two entries attributed to d18:2/24:0, and it’s likely that one of these is erroneous. The first assignment for GlcCer is also not correct, and including information on ionization (e.g., that these ions are likely [M-H2O+H]+ adducts) would be helpful for readers.

Why was acetate used instead of formate in negative ion mode?

Moving through the text, the nomenclature for SM and ceramides appears unchanged from the previous version, despite responses indicating otherwise.

Perhaps I am mistaken, but the Agilent 6460 is a triple quadrupole system, not a q-ToF.

If including screenshots of the file for images in the article, it would be helpful to disable spell check or remove the red underlines, perhaps with a simple tool like Paint (e.g., in Figure S3).

In Table 1, please replace "Hydrophobic chain composition" with "Attribution."

For the caption of Table S3, change "parent ion" to "product ions."

Author Response

Reviewer 2

While I find the exploration of microgravity intriguing, unfortunately, the manuscript contains experimental errors and issues in the interpretation of lipidomic data that prevent me from accepting the work.

My main concern derives from the answer to one of my previous questions.

The score plot in Figure S1 shows a QC sample entirely separated from the others, raising concerns. With such a score plot, it seems that analytical variability exceeds biological variability. If valid reasons exist for excluding that QC sample, the data should be re-evaluated, potentially lowering the acceptance criterion for variables from 40 to 20%. Removing this QC, which hopefully is due to the first chromatographic run, a good QC clustering has been obtained, and thus it is unclear why the 40% threshold has selected.

We thank the reviewer for raising this point. Upon reviewing the data, we discovered that a quality control (QC) sample, entirely distinct from the others, analyzed at the beginning of the sample run list, leading to a significant deviation from the remaining samples (see the old PCA -Figure S1). To address this, we removed the initial QC sample and replaced it with a QC sample randomly injected among the other samples, which yielded improved results (see the new PCA- Figure S1). Additionally, we lowered, as suggested by the reviewer, the feature threshold filter to 20%. As expected, we observed a decrease in the number of lipids/features detected. We reported this info in the text reporting the number of each lipid annonated (see row 211-213). Furthermore, this process did not impact the list of discriminating lipids which were not affected.

Could the authors clarify if data were autoscaled prior to PCA?

Before conducting the PCA, the data were autoscaled (by UV scaling ) by centering the data (subtracting the mean of each variable) and scaled dividing by the standard deviation. This information was added in Materials and Method section 2.9.

According to the authors’ responses, it appears that QC samples were only injected at the start of the run to condition the system, rather than throughout the analysis. If so, this approach does not align with standard metabolomics practices. Indeed, QCs should be injected between samples analyses to ensure the stability of the system.

We apologize for not specifying earlier: QC samples were included both at the beginning of the run to assess the conditioning of the chromatographic system and subsequently random injected between samples along with the lipid standard mixture. The latter were also injected randomly.

I must reiterate that an acceptance criterion this broad is not suitable for robust analysis.

Table 1 contains multiple nomenclature inconsistencies. For instance, the symbol ";" is used to separate chains in PE and SM, which is unconventional; either "/" or "_" should be used depending on whether regiochemistry is known. The nomenclature for sphingomyelins follows the older system, while ceramides use the more current one. A consistent approach is necessary.

We apologize for incongruences in the lipid nomenclature. We corrected and harmonized lipid nomenclature in text, Tables 1, 2, and 3 as well as in the Supplementary materials.

Additionally, we mistakenly reported in the first revision phosphatidylcholines as hydroxylated (see Table 1). We went back to the mass spec data and we confirm that we annotated phosphatidylcholines as PC (16:0/16:1) and not PC (16.0/18:1;O).

Additionally, some chromatographic assignments appear unusual. For example, PC 18:0/18:2;O elutes at 4.01 minutes, while PC 18:0/18:3;O elutes at 5.39 minutes. In general, increasing the number of double bonds should decrease retention time in RP chromatography. Are the authors confident in these assignments? I would expect that other, potentially more abundant lipids were also identified, and comparing these results with existing data could be informative.

We apologize again for incongruences in the lipid retention time of PC(18:0/18:3) and PC(18:0/18:2). After checking the chromatographic runs we reported the correct retention time which is now 4.86 min and 5.03, respectively.  We also re-check on the chromatograms retention times of each lipid and harmonized lipids rt in all tables.

A detailed review of data and terminology is needed, as the nomenclature in the table differs from that in the text. For PC lipids, the acyl chains are separated by ";" in the text, which appears to correspond to "_" in the authors' nomenclature.

We replaced ";" with "/" in the text for the known sn-position of acyl/alkyl constituents.

In the table, for PC, the second acyl chain is consistently hydroxylated, and regiochemistry is inferred.

As stated before we mistakenly reported in the first revision phosphatidylcholines as hydroxylated (see Table 1). We got back to the mass spec data and we confirm that we annotated phosphatidylcholines as PC(16:0/16:1) and not PC (16.0/18:1;O).

This goes beyond a simple nomenclature error. Although readers may infer from m/z values that PCs are not hydroxylated, the current wording could mislead non-experts familiar with the nomenclature system. They might interpret it as suggesting that hydroxylation occurs selectively during microgravity experiments, consistently on the sn-2 chain, potentially implying stereoselective processes. Therefore, careful attention to this aspect is essential!

Again, we are sorry for the mistake, we understand the importance of the point raised of the reviewer. We believe to have made all the corrections to the text and tables needed.

Some of the MS/MS data provided in the supplementary material may not be sufficiently informative. For instance, the water loss observed in certain Cer does not offer additional insights; it might be more beneficial to include the signal for the sphingoid base as you have done for other cases, even if this is not likely the base peak. Some assignments also seem potentially incorrect. For example, in positive ion mode, SM typically form m/z 184 as the base peak. You have two entries attributed to d18:2/24:0, and it’s likely that one of these is erroneous. The first assignment for GlcCer is also not correct, and including information on ionization (e.g., that these ions are likely [M-H2O+H]+ adducts) would be helpful for readers.

For the MS/MS analysis of ceramides we added the qualitative ion used along the quantitative ion. We originally reported SM 18:2/24:0 in two separate rows; however, we have now consolidated this information into a single row, where both fragment ions are reported together. Additionally, in Table S3, due to a layout error, Glc-Cer 18:2;25:0 should be interpreted as SM(18:2/25:0). We changed the text accordingly.

Why was acetate used instead of formate in negative ion mode?

In our lipidomic analysis conducted in negative ion mode we prefer to use acetate because provides a cleaner spectrum with fewer interfering peaks compared to formate. Another reason of using acetate is related to the minimizing matrix effects.

Moving through the text, the nomenclature for SM and ceramides appears unchanged from the previous version, despite responses indicating otherwise.

We apologize again, we have revised and corrected the text and Tables. We changed the text accordingly.

Perhaps I am mistaken, but the Agilent 6460 is a triple quadrupole system, not a q-ToF.

We apologize for the mistake; the correct model number is Agilent UHPLC-IM-QTOF/MS 6560.

If including screenshots of the file for images in the article, it would be helpful to disable spell check or remove the red underlines, perhaps with a simple tool like Paint (e.g., in Figure S3).

We have revised Figure S3 as suggested.

In Table 1, please replace "Hydrophobic chain composition" with "Attribution."

We changed the Table accordingly.

For the caption of Table S3, change "parent ion" to "product ions."

We have changed the wording of Table S3 as suggested.